# Health Benefits Quantification for New-Energy Vehicles Promotion: A Case Study of Beijing

**DOI:** 10.3390/ijerph192113876

**Published:** 2022-10-25

**Authors:** Yue Wang, Yang Wen, Yingying Xu, Lei Shi, Xuan Yang

**Affiliations:** 1School of Environment and Natural Resources, Renmin University of China, Beijing 100872, China; 2Chinese Academy of Macroeconomic Research, Beijing 100038, China; 3Institute of Spatial Planning & Regional Economy, National Development and Reform Commission, Beijing 100038, China

**Keywords:** new-energy vehicles, health effects, cost-benefit analysis, Beijing

## Abstract

Considering that the promotion of new-energy vehicles (NEVs) is a potential measure to address urban air pollution, the Chinese government has launched subsidy schemes to improve its market penetration. The purpose of this study is to quantify the economic benefits of NEV promotion from 2016 to 2019 and compare them with the cost of government subsidies in Beijing, so the effectiveness of the NEV policies can be evaluated and valuable recommendations can be provided for decision-making. The exposure–response model and the cost of illness approach was applied to evaluate the health and economic benefits of NEV promotion. Our results are as follows: (1) promoting NEVs can reduce the PM_2.5_ concentration significantly, the average annual concentration reduction was between 3.23 μg/m^3^ and 4.61 μg/m^3^; (2) at least 37,545 illnesses and deaths in Beijing could be reduced through NEV promotion annually, internal disease (15–64) was the most affected illness; (3) the economic benefits of NEV promotion were far more than the cost and the net benefits stock reached 33.71 billion CNY in 2019, accounting for 0.95% of Beijing’s GDP in the same period. This study provides references in the perspective of environmental economics for the policy formulation of promoting NEVs.

## 1. Introduction

With the rapid development of the economy and advancement of urbanization, motor vehicles have brought great convenience to residents, but they also became one of the major pollution sources [1]. In China, total motor vehicle emissions containing particulate matter, total hydrocarbon and other pollutants exceed approximately 44 million tons per year [2]. Local source emissions accounted for two-thirds of Beijing’s annual primary sources of PM_2.5_, while mobile source emissions contributed the most toward local source emissions in 2017 [3]. By types of vehicles, bus particulate matter emissions made up 9.9% of all vehicle emissions in 2019, while truck particulate matter emissions totaled 62,000 tons, or 90.1% of all vehicle emissions. For different types of fuels, gasoline cars produced 5.576 million, 1.328 million and 0.303 million tons of CO, HC and NO_x_ emissions in 2019, respectively, accounting for 80.3%, 77.5% and 4.9% of all vehicle emissions nationally. Diesel vehicle emissions of the aforementioned pollutants were 1.286 million, 0.213 million, 5.532 million and 0.069 million tons, or 18.5%, 12.5%, 88.9% and 99%, respectively. Gas vehicle emissions of CO, HC and NO_x_ were 81,000, 171,000, and 387,000 tons, or 1.2%, 10.0%, and 6.2%, respectively [4]. These emissions are the main causes of haze and photochemical smog pollution [5], which seriously threaten the atmospheric environment and human health [6,7]. The consumption of new-energy vehicles (NEVs) is considered as a potential measure to alleviate urban air pollution [8]. Battery electric vehicles, plug-in hybrid electric vehicles and fuel cell electric vehicles are the most common NEVs in China [9]. Compared to conventional vehicles, NEVs consume fewer fossil fuels and directly reduce the emissions of greenhouse gases as well as atmospheric pollutants during the driving cycle [10]. The demand for innovation, the high original cost of production and market uncertainty [11,12] has made NEV promotion take more risks than traditional vehicle promotion [13]. Therefore, the Chinese government has launched subsidy schemes to improve the market penetration of NEVs [14]. In 2013, the subsidy policy began to push the NEVs industry to make further progress [15], and China’s NEVs started to be mass-produced in 2014.

As one of the major markets for NEVs in China, Beijing was ranked first in terms of the adoption of approximately 23,500 battery electric vehicles in 2015, accounting for 9.5% of the total adoption of battery electric vehicles in the country. Since the number of NEVs in Beijing has grown rapidly, in order to control the total number of vehicles in the city, Beijing has adopted a quota lottery policy for not only gasoline vehicles (GVs) but also NEVs, which refers to a policy design where the government has a limited quota of the number of annual vehicles, and consumers enter a lottery to decide who will be able to get a vehicle [16]. Every year, the Beijing government will issue a certain number of vehicle purchase quotas according to the road traffic conditions, parking space supply and environmental capacity. If a person wants to buy NEVs in Beijing, they need to submit an application first and wait in line. When the number of quotas released by the Beijing government exceeds the number of people in front of them, they will be allowed to make the purchase. In 2016, the Ministry of Industry and Information Technology of China began to release the liquidation results of NEV promotion and application subsidies year by year. The detailed data provide a basis for comparing the cost of subsidies with the benefits they generate. Therefore, Beijing is an ideal area for NEV research due to its detailed statistical data and the regularity of quota policies. As shown in Figure 1, the ownership of NEVs in Beijing had grown rapidly from 2014 to 2016; the peak of annual growth rate reached 272% during this period. After 2016, Beijing’s quota for NEVs had been kept at 60,000 per year, which caused a significant decline in annual growth rate. Obviously, while encouraging the development of NEVs, Beijing is facing the pressure of controlling the total number of vehicles. A better understanding of the effect of NEVs in Beijing would help policymakers to make efficient policies at both local and national levels.

Whether the promotion of NEVs can produce the expected benefits [17,18], and whether the government subsidies are efficient [19] are the issues that have drawn wide concern. Many studies have focused on these issues. Zhang et al. proposed an evaluation index to measure the effectiveness of the two subsidies in reducing emissions of atmospheric pollutants among different NEVs. They found that by prioritizing the promotion of NEVs with higher comprehensive subsidy efficiency, one can achieve greater reduction in the emissions of atmospheric pollutants with equal financial funds [3]. Wang et al. presented a well-to-wheel (WTW) analysis to estimate the WTW fossil fuel consumption, greenhouse gas emissions and air pollutants emissions of VOCs, CO, NO_x_, SO_x_, PM_2.5_ and PM_10_ for hydrogen fuel cell vehicles (HFCVs) under 12 hydrogen pathways in China. They found that HFCVs, based on all hydrogen pathways, can achieve a significant reduction in VOCs and CO emissions on a WTW basis, in comparison with gasoline-fueled internal combustion engine vehicle in 2017 [20]. Moreover, Wang et al. took advantage of a promotion project in China as a natural experiment, and explored the causal effect of promoting NEVs on reducing carbon intensity. They found the promotion of NEVs reduced the carbon intensity in demonstration cities by about 4.5% [21]. Based on the dataset of Chinese NEV firms’ financial performance from 2013 to 2017, Yu et al. demonstrated the government subsidies can exert impacts on the financial performance of Chinese enterprises of NEVs utilizing panel regression models [22].

Moreover, adding the comparison of health effect attributes and policy costs seems to provide a wider context of the assessed policies [23,24,25]. Based on the case study of Thessaloniki, Sarigiannis et al. (2017) computed the potential co-benefits from reduced greenhouse gas emissions on public health by the year 2020 utilizing state-of-the-art concentration response functions for particulate matter, NO_2_ and C_6_H_6_. Results showed that the promotion of electric vehicles would provide monetary savings from the reduction in PM_10_, PM_2.5_, NO_2_ and C_6_H_6_ exposure up to 60.4, 49.1, 41.2 and 1.08 million Euros [26]. Raifman et al. (2021) analyzed all 378 counties in the Transportation Climate Initiative (TCI) region and found that even for the scenario with the smallest investment in active mobility, when it is fully implemented, TCI would result in hundreds of fewer deaths per year across the region, with monetized benefits in the billions of dollars annually [27]. Coomes et al. (2022) estimated benefits across a suite of child health outcomes in 42 New York City neighborhoods under the proposed regional Transportation and Climate Initiative. Results showed that a cap-and-invest strategy to reduce carbon emissions from the transportation sector could provide substantial health and monetized benefits to children in New York City through reductions in criteria pollutant concentrations, and with greater benefits among Black and Hispanic children [28].

The results of exciting research provides important information regarding the effectiveness of subsidy policies in the NEVs industry. However, most of the existing research remains focused on the measurement of pollutant emissions, lacking the quantitative evaluation of economic benefits, and thus unable to analyze the cost-benefit on the basis of monetization.

The purpose of this study is to quantify the economic benefits of NEV promotion from 2016 to 2019 and compare them with the cost of government subsidies in Beijing, so the effectiveness of the NEV policies can be evaluated and valuable recommendations can be provided for the decision-making surrounding NEV promotion and the pollutant emission treatment. The remainder of this paper is organized as follows: Section 2 elaborates methodology and data. The empirical results and corresponding findings are provided in Section 3. The conclusions and policy implications are presented in Section 4.

## 2. Materials and Methods

Atmospheric particulate pollution had become the fourth major pathogenic factor for Chinese people, after stroke, ischemic heart disease, and chronic obstructive pulmonary disease [29]. Significantly, PM_2.5_ is now the leading environmental contributor to the global burden of disease [30]. Compared with other air pollutants, PM_2.5_ is a more suitable research object for health benefit quantification due to its severe damage and mature research system. Therefore, PM_2.5_ is selected as the target pollutant in the study.

Light-duty gasoline-fueled passenger vehicles (LDGVs) are the most common type of vehicle in Beijing. All out-of-city diesel vehicles with emissions standards lower than China III were prohibited from travelling within the sixth ring road as of September 2017 [31]. According to Beijing Transport Development Annual Report in 2020, the ownership of LDGVs in 2019 accounted for 85.8% of the ownership of total motor vehicles. Moreover, for the lack of travel distance and emission factors information of other types of vehicles in the public data of Beijing, the LDGVs were chosen as the research object in this study. Assuming that NEVs do not emit pollutants, the emission reduction of NEVs can be seen as the emissions of the GVs they replace. There are two paths for GVs to produce PM_2.5_, one is direct emission, and the other is the secondary conversion of other pollutants. The estimation method of PM_2.5_ emission is shown in Formula (1).
(1)$$E_{PM_{2.5}}^{t}=N^{t}\cdot d^{t}\left(\delta^{t}{}_{PM_{2.5}}+\delta^{t}{}_{NOx}\cdot NOR\right)$$
where, $E_{PM_{2.5}}$ denotes the PM_2.5_ emission, *t* denotes the periods. N denotes the number of the NEVs, and the number of quota and stock are considered separately. d is the annual travel distance of LDGVs. The number of NEVs and the travel distance data are gathered from Beijing Transport Development Annual Report (from 2017 to 2020). $\delta_{PM_{2.5}}$ and $\delta_{NOx}$, respectively represent the direct emission factor of PM_2.5_ and NO_x_. Based on the calculation results of existing research [32], this research obtained the vehicle emission factors of NO_x_ and PM_2.5_ in Beijing from 2016 to 2019 by interpolation method, as shown in Figure 2. To evaluate the differences between real emissions and the test results of approval, the estimation of emissions factors considered various test cycles including the legislative driving cycle and alternative driving cycles with more real-world features. The estimation also considered influence from the vehicle technology group, accumulated vehicle kilometers traveled, average speed, fuel quality, and environmental conditions. NOR represents the nitrogen oxidation ratio, which is generally adopted to represent the conversion ratio of NO_x_ to PM_2.5_, according to the practice of Xia et al. [33], NOR is set as 0.25 in this study.

Environmental epidemiology and toxicology studies have confirmed that exposure to polluted air can cause respiratory, cardiovascular and other diseases, and even premature death [34,35,36,37]. The health benefits generated by the promotion of NEVs are taken as the evaluation index of the economic benefits. According to the existing research results, this paper selects diseases that have been confirmed to be associated with PM_2.5_ pollution and have detailed statistics, including disease of respiratory system, cardiovascular disease, pediatric diseases (0–14), internal disease (15–64), chronic bronchitis, acute bronchitis, asthma and death to evaluate the health effects of NEV promotion. The exposure–response model of PM_2.5_ for each health endpoint, which were derived from available epidemiologic studies and based on a relative risk model in the form of Poisson regression, has become a mature theory for measuring health effects of pollution [38] and were used to quantify the health effects of particulate air pollution [39,40,41,42]. This study applied the exposure–response model to estimate the health impact of emission reduction caused by NEVs promotion, as shown in Formula (2).
(2)ΔRR=P·RR0·[1−1exp(β·ΔC)]
where, ΔRR denotes the change of residents’ health effect caused by PM_2.5_. *P* is the exposed population. Considering that more than 75% of the population in Beijing lives within the perimeter of the Sixth Ring Road, these people are closer to the urban center and more affected by traffic air pollution, so, the number of permanent residents live within the Sixth Ring Road of Beijing is selected as the exposed population. $RR_{0}$ represents the disease incidence in general conditions. β is the exposure–response indicator which represents the percentage increase in population morbidity and mortality caused by a 1µg/m^3^ increase in pollutant concentration. ΔC denotes the change in PM_2.5_ concentration caused by vehicle exhaust emissions. Data supporting the exposure-response model were obtained from Beijing Statistical Yearbook and existing research [43], as shown in Table 1.

Based on that, the cost of illness approach is used to monetize the health effects. Scholars have estimated the economic losses per unit of death and various illness in Beijing [44,45,46]. On the basis of these research results, this paper uses consumer price index to convert economic losses into current year prices from 2016 to 2019, as shown in Figure 3.

## 3. Results and Discussion

### 3.1. The Emission Reduction Effects of NEV Promotion

The PM_2.5_ emission reduction of NEVs in Beijing was estimated by Formula (1). The effect increments were calculated with the quota of NEVs over the years, and the effect stocks were calculated with the annual retain number of NEVs. To prepare for the subsequent economic benefit analysis, the emission reduction needs to be converted into concentration change. According to a publicly released information by Atmospheric Department of Beijing Environmental Protection Bureau, motor vehicles are the largest source of pollution in Beijing. In early 2017, the last large coal-fired power plant in Beijing was shut down, and the goal of utilizing coal-free power plants has been basically realized in Beijing during the study period from 2016 to 2019. Therefore, the proportion of stationary sources in Beijing had been reduced to a very low level and has less effect on the background concentration of PM_2.5_. While the geographical scope of this research is barely large enough to be significantly different in spatial parameters, this study assumed that the climatic and meteorological conditions and other spatial parameters in this area are the same. While the concentration of PM_2.5_ can be greatly affected by the large difference in climatic conditions between heating season, the period when government supply heating for the residents, and non-heating season in Beijing, the two periods were considered separately. According to existing results, the average height of the static weather boundary in heating season and non-heating season in Beijing is 420 m and 600 m, respectively. This paper focused on vehicle exhaust within the sixth ring road of Beijing, the area of this range is 2267 km^2^. Hence, under the static stability weather, Beijing can be approximately regarded as a box with volume of 9.52 × 10^11^ m^3^ in heating season and 1.36 × 10^12^ m^3^ in non-heating season [47,48]. The decrease of PM_2.5_ concentration driven by NEV promotion is shown in Figure 4.

As the quota for NEVs in Beijing from 2016 to 2019 remained unchanged, the effect of increments of the PM_2.5_ concentration reduction caused by NEV promotion in Beijing had not changed significantly. The average annual concentration reduction increments were between 3.23 μg/m^3^ and 4.61 μg/m^3^. As for the effect stocks, there was a significant decline between 2017 and 2018 although the ownership of NEVs in Beijing kept growing during this period; see Figure 1. One of the reasons is that emission factors of PM_2.5_ and NO_x_ continued to decline with the development of mitigation technologies. The other and more important reason is the apparent decline in travel distance from 2017 to 2018; see Figure 2. In 2019, the concentration reduction effect reached 14.40 μg/m^3^ to 20.57 μg/m^3^, accounting for 34.29% to 48.98% of Beijing’s annual average PM_2.5_ concentration (42 μg/m^3^) at the same period. This result shows the significant PM_2.5_ emission reduction effects of promoting NEVs.

### 3.2. The Economic Effects of NEV Promotion

To monetize the emission reduction effects of NEV promotion, its health benefits should be calculated firstly. The results estimated through Formula (2) are shown in Figure 5. The concentration of PM_2.5_ in heating season is relatively high, so it was taken as the limes superior, and that in non-heating season was considered as the limes inferior, so as to get the estimation interval of the effects. This paper did not take energy source emissions into account when measuring health benefits in Beijing. According to the Energy Balance Table of Beijing from 2016 to 2019, almost (97.55%, 100%) of coal, 100% of natural gas, 100% of heat and (97.99%, 98.95%) of electricity in Beijing were imported from other provinces. Therefore, most of Beijing’s motor vehicle energy source emissions are in other provinces, which has very little impact on the health of people living in Beijing.

The health benefits of NEV promotion in Beijing is obvious, as is shown in Figure 5, NEV promotion can reduce the total number of illnesses and deaths in Beijing by at least 37,545 annually in terms of the effect increments, and the limes superior reached 89,292 in 2017. As for the effect stocks, the total number was only within the range of (72,353, 102,032) in 2016, then it reached the range of (203,531, 284,953) in 2019, which increased by (181.30%, 179.28%). Internal disease (15–64), acute bronchitis and pediatric diseases (0–14) were the top three affected diseases, and the impact of NEV promotion on them were far more significant than other diseases. Among them, internal disease (15–64) was the most affected one, the number of people suffering from these diseases could drop by 17,318 to 24,513 per year due to the promotion of NEVs, and the effect stocks reached the range of (77,098, 109,007), accounting for 37.88% to 38.25% of the total affected number. Based on these results, the monetization of NEV promotion effects was calculated, as shown in Figure 6.

The promotion of NEVs were mostly economically beneficial for the reduction of chronic bronchitis due to the high medical cost of this illness. Due to NEV promotion, the economic loss of chronic bronchitis could be lowered at least 4.4 billion CNY annually, and the limes superior reached 9.7 billion CNY in 2017. It was followed by death, at least 2.1 billion CNY in economic losses could be avoided per year, and the peak of limes superior reached 4.7 billion CNY. The combined economic benefits of NEV promotion on chronic bronchitis and death accounts for over 98% of the total annually. As for the effect stocks, after fluctuating upward from 2016 to 2019, the total effects of NEV promotion reached the range of 37.4-51.8 billion CNY in 2019, accounting for 10.57% to 14.63% of Beijing’s GDP at the same period. Therefore, the promotion of NEVs can not only be conducive to the emission of air pollutants, but will also produce huge social and economic benefits, which is consistent with the verdict of Gai et al. [49].

### 3.3. The Cost-Benefit Analysis of NEV Promotion

A cost-benefit analysis can be used to assess emission reduction measures [50]. Wang et al. (2020) used detailed modelling of energy system transformation, cross-sectoral connectivity and technology penetration to quantify the associated health co-benefits from reduced co-emitted air pollutants. Their results showed that the annualized monetary benefits of health co-benefits (US$215 billion) exceeded the greenhouse gasses abatement cost (US$106 billion) by US$109 billion [51]. A cost-benefit analysis of emission control measures is an important part of the development and refinement of policies in the transportation sector, especially in China [52,53]. Shih and Tseng (2014) developed an air resource co-benefits model to estimate the social benefits of a sustainable energy policy and averted years of life lost, which found that the benefit-cost ratio of 1.9–2.1 under the energy efficiency improvements scenario was lower than the benefit-cost-ratio of 7.2–7.9 under the renewable energy scenario [54]. Cost-benefit analysis, as a tool for evaluating policies, can assist decision-makers in balancing the costs and benefits of public policies. In this study, the comparison of the economic benefits of NEV promotion from 2016 to 2019 and the cost of government subsidies in Beijing is performed by the cost-benefit analysis to develop more effective and economic measures in the future.

Government subsidies for NEVs in Beijing were used to quantify the cost of promoting NEVs. Since subsidies cannot represent the full cost of NEV promotion by government, the cost is underestimated in this study. The subsidy funds for the promotion and application of NEVs need to go through multiple rounds of liquidation and audit, so, the actual government subsidy issued could not be consistent with the number of vehicle purchases. In this paper, the funds in the final announcement published on the official website of the Ministry of Industry and Information Technology of the People’s Republic of China were used as the cost data. The aggregate benefits of all cases and their comparison with the costs over the years are shown in Table 2.

Obviously, the benefits of NEV promotion far outweigh the costs. Conservatively estimated, the incremental net benefits of NEV promotion over the years were at least 6.40, 9.82, 5.86 and 5.92 billion CNY, accounting for 0.26%, 0.35%, 0.19% and 0.17% of Beijing’s GDP at the same period. From the perspective of stock, the net benefits were at least 10.65, 26.90, 22.27 and 33.71 billion CNY, accounting for 0.43%, 0.96%, 0.73% and 0.95% of Beijing’s GDP in the corresponding years. Therefore, within the scope of this study, the promotion of NEVs in Beijing was effective and could provide great investment opportunities in the future.

### 3.4. Limitations

This study limited the scope to Beijing which has little local energy supply and relies mainly on energy from other provinces. As for the primary energy source, Beijing imported 97.99–98.95% of its electricity from other provinces according to the Energy Balance Table of Beijing from 2016 to 2019. Therefore, most of Beijing’s motor vehicle energy source emissions are in other provinces. Although these emissions have very little impact on the health of local people in Beijing, provinces that supply electricity to Beijing may face local air pollution problems. However, due to the lack of public data, it is difficult to identify the provinces that supply electricity to Beijing. The proportions of energy sources (coal, hydro, solar, wind, etc.) generating the electricity being delivered to Beijing are hard to identify as well, which limits further refinement of the study. With the continuous refinement of public data, the negative impacts by electricity generation outside Beijing can be considered in the future.

## 4. Conclusions

Based upon the above analysis, some findings can be integrated. Firstly, promoting NEVs can reduce the PM_2.5_ concentration significantly; the average annual concentration reduction increments from 2016 to 2019 was between 3.23 μg/m^3^ and 4.61 μg/m^3^, and the effect stocks in 2019 reached 34.29% to 48.98% of Beijing’s annual average PM_2.5_ concentration. These conclusions that NEV promotion benefits the environment are in line with the findings of Wang et al. (2021) [55]. However, Wang et al. (2021) [55] estimated the influence of the promotion of NEVs during a full lockdown in 2019 due to COVID-19, this study analyzed a longer time period from 2016 to 2019. Secondly, the emission reduction of NEVs has produced huge social benefits, which can be reflected in the impact on residents’ health. At least 37,545 illnesses and deaths in Beijing could be reduced through NEV promotion annually, and the number of people suffering from internal disease (15–64) could drop by at least 17,318 per year, which is consistent with the conclusion of Liang et al. (2019) [56]. This study also investigated how the number of persons with internal disease (15–64) might be decreased by promoting NEVs, in contrast to Liang et al. (2019) [56] who only examined how many annual premature deaths could be decreased by promoting NEVs. Finally, the health effects of NEV promotion will generate significant economic benefits that can greatly outweight the cost. In 2019, the net benefits stock reached 33.71 billion CNY, accounting for 0.95% of Beijing’s GDP.

Recently, Beijing is gradually reducing subsidies for NEVs and it has become the first city in China to implement subsidy retrogression. In 2018, Beijing’s subsidy for electric vehicles was no longer 1:1 to the central government’s, the municipal subsidy had been adjusted to 50% of the central subsidy. In 2019, Beijing had decided to stop subsidizing electric vehicles, the subsidy for fuel-cell vehicle had been changed to 50% of the central subsidy. However, the results of this study showed that policies to promote NEVs can produce environmental benefits, social benefits and economic benefits that could be several times the cost of subsidies. In reality, the subsidy reduction is a decision made after comprehensive consideration of multiple factors. The results of this study only provide reference in the perspective of environmental economics for the policy formulation of promoting NEVs, which shows that the promotion of NEVs is efficient in terms of both atmospheric environmental protection and economic development.

## Figures and Tables

**Figure 1 ijerph-19-13876-f001:**
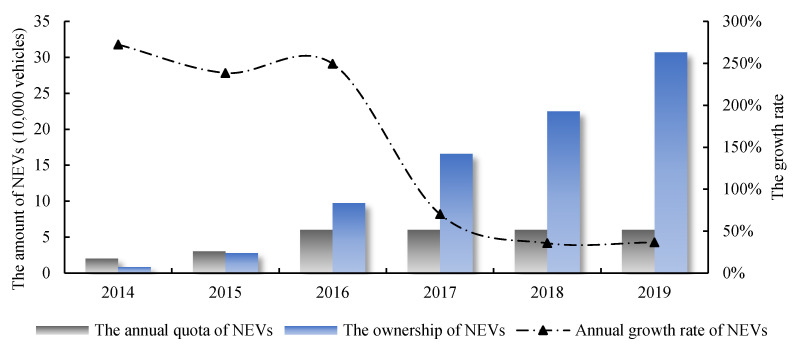
Quota, ownership and growth rate of NEVs in Beijing from 2014 to 2019.

**Figure 2 ijerph-19-13876-f002:**
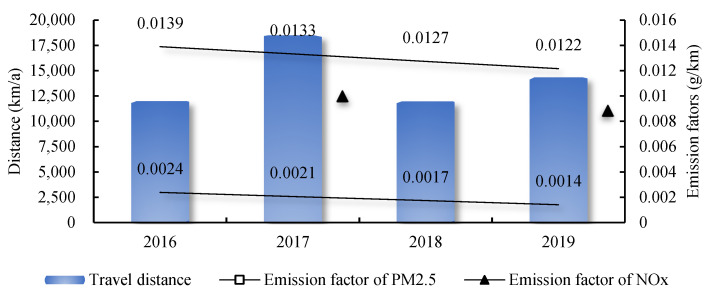
Travel distance and emission factors of LDGVs in Beijing from 2016 to 2019.

**Figure 3 ijerph-19-13876-f003:**
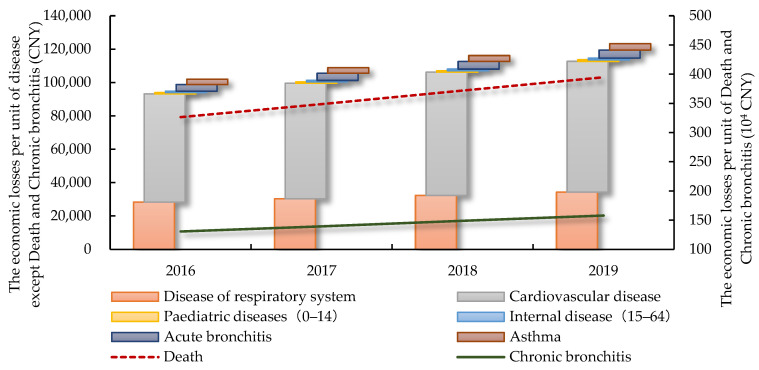
The economic losses per unit of death and various illness in Beijing from 2016 to 2019.

**Figure 4 ijerph-19-13876-f004:**
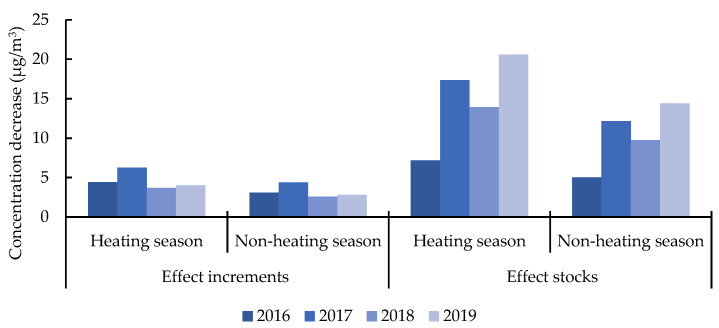
The decrease of PM_2.5_ concentration driven by NEV promotion in Beijing from 2016 to 2019.

**Figure 5 ijerph-19-13876-f005:**
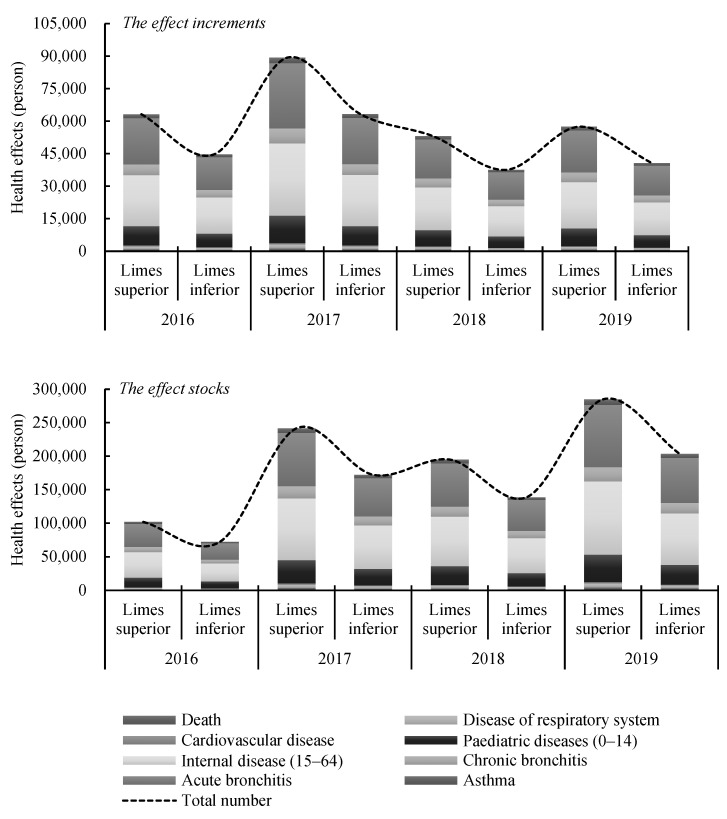
The health effects of NEV promotion in Beijing from 2016 to 2019.

**Figure 6 ijerph-19-13876-f006:**
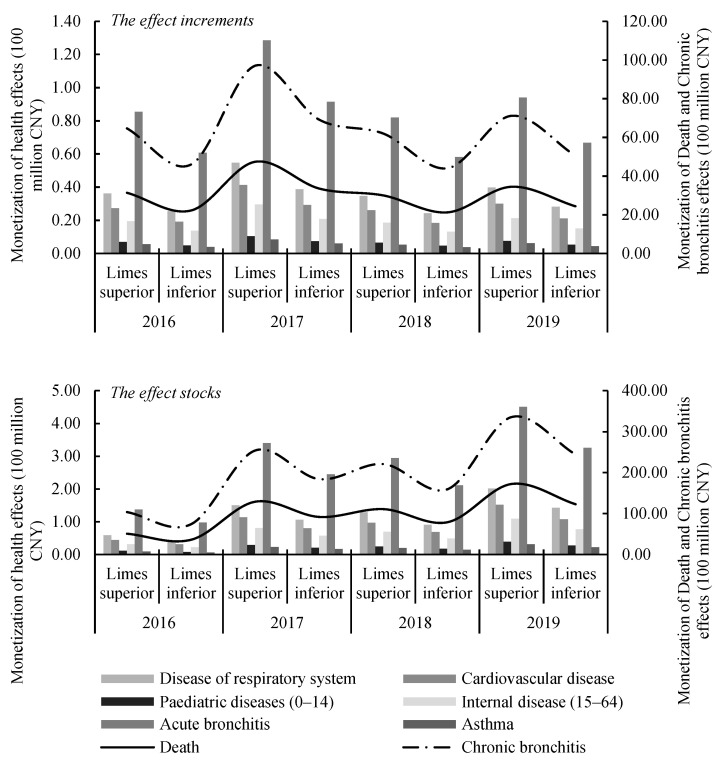
Monetization of NEV promotion effects in Beijing from 2016 to 2019.

**Table 1 ijerph-19-13876-t001:** Data of variables in exposure-response model.

Variable	Unit		Value
*P*	10^4^ people	2016	1649.23
		2017	1647.56
		2018	1635.04
		2019	1634.58
$RR_{0}$	%	Death	0.45%
		Disease of respiratory system	1.62%
		Cardiovascular disease	0.86%
		Pediatric diseases (0–14)	22.04%
		Internal disease (15–64)	66.55%
		Chronic bronchitis	0.69%
		Acute bronchitis	3.80%
		Asthma	1.19%
β	%	Death	0.30%
		Disease of respiratory system	0.11%
		Cardiovascular disease	0.07%
		Pediatric diseases (0–14)	0.06%
		Internal disease (15–64)	0.05%
		Chronic bronchitis	1.01%
		Acute bronchitis	0.79%
		Asthma	0.21%

**Table 2 ijerph-19-13876-t002:** The Cost-Benefit of NEV promotion in Beijing from 2016 to 2019. (Unit: 100 million CNY).

	Increments			Stocks		
	Cost	Benefit		Cost	Benefit	
		Limes Superior	Limes Inferior		Limes Superior	Limes Inferior
2016	5.41	97.84	69.45	5.41	157.17	111.91
2017	6.46	147.15	104.67	11.87	390.08	280.88
2018	7.90	93.74	66.48	19.77	337.97	242.45
2019	17.11	107.60	76.34	36.88	517.50	373.93

## Data Availability

The data are not publicly available due to privacy.

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
