# Peer review of "Health Benefits Quantification for New-Energy Vehicles Promotion: A Case Study of Beijing"

_ijerph, 2022, doi:10.3390/ijerph192113876_

Round 1
Reviewer 1 Report
Dear authors,
The article seems very interesting to me. The conclusions show that the efficiency of the measures both from an economical and healthy point of view. It is very disappointing to see that theses measures are receding in Beijing area.
My comments are related to the methodology section.
Formula (2) in line 132 is used to apply the exposure-response model. Nothing is explained about this model and only reference [26] is proposed. This model must be explained, and some more references presented.
Something similar occurs with the cost-benefit analysis. Some references must be included showing some examples where cost benefits are used in health studies. And also, a brief description of how to carry a cost-benefit analysis.
Best regards
Author Response
Dear Reviewer:
Thank you for reading our manuscript and reviewing it, which helped us improve it to a better scientific level. Those comments are all valuable and very helpful for revising and improving our paper, as well as the important guiding significance to our researches. We have carefully revised our manuscript and rechecked the entire text for errors in English spelling and grammar. The changes were marked up using the “Track Changes” function in the revised manuscript and highlighted in red in the response letter. We hope that the revision is acceptable and look forward to hearing from you soon.
Once again, thank you very much for your detailed comments and suggestions. Should there been any other corrections we could make, please feel free to contact us.
By the way, the COVID-19 pandemic is directly or indirectly affecting each one of us as members of the global community. We hope you and your loved ones continue to take the necessary precautions and stay safe and healthy.
Best Regards,
Yue Wang, Yang Wen, Yingying Xu, Lei Shi and Xuan Yang*
2022/10/17
Response to Reviewer 1 Comments
Point 1: Formula (2) in line 132 is used to apply the exposure-response model. Nothing is explained about this model and only reference [26] is proposed. This model must be explained, and some more references presented.
Response 1: Thanks for your valuable comments and suggestions. This is a great guidance to our research work and is very helpful for us to improve the academic quality of this manuscript. For your valuable suggestions, we have detailed the explanation of exposure-response model. Moreover, we have complemented related reference to promote the rationality of utilizing this model. The relevant statements are as follows. Thank you very much again for your constructive comments.
The exposure–response model of PM2.5 for each health endpoint, which were derived from available epidemiologic studies and based on a relative risk model in the form of Poisson regression, has become a mature theory for measuring health effects of pollution [36] and were used to quantify the health effects of particulate air pollution [37-39]. This study applied the exposure–response model to estimate the health impact of emission reduction caused by NEVs promotion, as shown in formula (2).
References:
Kan, H.D., Chen, B.H. Particulate air pollution in urban areas of Shanghai, China: health-based economic assessment. Sci. Total Environ. 2004, 322: 71-79.
Zhao, N., Li, B., Li, H., Ahmad, R., Peng, K., Chen, D., Yu, X., Zhou, Y., Dong, R., Wang, H., Ju, X., Zayan, A.M.I. Field-based measurements of natural gas burning in domestic wall-mounted gas stove and estimates of climate, health and economic benefits in rural Baoding and Langfang regions of Northern China. Atmos. Environ. 2020, 229: 117454.
Altieri, K.E., Keen, S.L. Public health benefits of reducing exposure to ambient fine particulate matter in South Africa. Sci. Total Environ. 2019, 684: 610-620.
Point 2: Something similar occurs with the cost-benefit analysis. Some references must be included showing some examples where cost benefits are used in health studies. And also, a brief description of how to carry a cost-benefit analysis.
Response 2: Thanks a lot for the suggestion, it is very useful for improving the scientificity of our research. To follow your valuable advice, we have included some references to show some examples where cost-benefit analysis is used in health studies. Additionally, we have supplemented the comprehensive description and interpretation to explain how to carry a cost-benefit analysis. Thanks again for your reasonable suggestions. The detailed statement is as follows.
Cost-benefit analysis can be used to assess emission reduction measures [47]. Wang et al. (2020) used detailed modelling of energy system transformation, cross-sectoral connectivity and technology penetration to quantify the associated health co-benefits from reduced co-emitted air pollutants. Results showed that the annualized monetary benefits of health co-benefits (US$215 billion) exceeded the Greenhouse Gasses abatement cost (US$106 billion) by US$109 billion [48]. Cost-benefit analysis of emission control measures is an important part of the development and refinement of policies in the transportation sector, especially in China [49,50]. Shih and Tseng (2014) developed an Air Resource Co-benefits model to estimate the social benefits of a Sustainable Energy Policy and averted years of life lost, which found that the benefit-cost ratio of 1.9-2.1 under the energy efficiency improvements scenario was lower than the benefit-cost-ratio of 7.2-7.9 under the renewable energy scenario [51]. Cost-benefit analysis, as a tool for evaluating policies, can assist decision-makers in balancing the costs and benefits of public policies. In this study, the comparison of the economic benefits of NEVs promotion from 2016 to 2019 and the cost of government subsidies in Beijing is performed by the cost-benefit analysis to develop more effective and economic measures in the future.
References:
Liu, Y., Liao, W., Li, L., Huang, Y., Xu, W., Zeng, X. Reduction measures for air pollutants and greenhouse gas in the transportation sector: A cost-benefit analysis. J. Clean. Prod. 2018, 207: 1023-1032.
Wang, T., Jiang, Z., Zhao, B., Gu, Y., Liou, K.N., Kalandiyur, N., Zhang, D., Zhu, Y. Health co-benefits of achieving sustainable net-zero greenhouse gas emissions in California. Nat. Sustain. 2020, 3: 597-605.
Bollen, J., van der Zwaan, B., Brink, C., Eerens, H. Local air pollution and global climate change: a combined cost-benefit analysis. Resour. Energy Econ. 2009, 31: 161-181.
Rosqvist, L.S., Hiselius, L.W. Online shopping habits and the potential for reductions in carbon dioxide emissions from passenger transport. J. Clean. Prod. 2016, 131: 163-169.
Shih, Y.H., Tseng, C.H. Cost-benefit analysis of sustainable energy development using life-cycle co-benefits assessment and the system dynamics approach. Appl. Energy 2014, 119: 57-66.
Special thanks for these constructive comments!
Reviewer 2 Report
Please find the attachment.

Author Response
Dear Reviewer:
Thank you for reading our manuscript and reviewing it, which helped us improve it to a better scientific level. Those comments are all valuable and very helpful for revising and improving our paper, as well as the important guiding significance to our researches. We have carefully revised our manuscript and rechecked the entire text for errors in English spelling and grammar. The changes were marked up using the “Track Changes” function in the revised manuscript and highlighted in red in the response letter. We hope that the revision is acceptable and look forward to hearing from you soon.
Once again, thank you very much for your detailed comments and suggestions. Should there been any other corrections we could make, please feel free to contact us.
By the way, the COVID-19 pandemic is directly or indirectly affecting each one of us as members of the global community. We hope you and your loved ones continue to take the necessary precautions and stay safe and healthy.
Best Regards,
Yue Wang, Yang Wen, Yingying Xu, Lei Shi and Xuan Yang*
2022/10/17

Reviewer 3 Report
1. Please provide crrent health costs (and/or DALY) by convenstional vehicles and the sheare of mobile sources (compared with fixed sources).
2. Please add references related to cost and benefit analysis of transport negative externalities in the US and Europe.
3. Please provide the travel distance and emission factors by vehicle type. How did you consider trucks and lollys? In addition, these are affected by engine size, fuel type, speed and ages are quite important factors. How did you considered in the analysis?
4. Not only mobile but also fixed emission sources affects the air pollution level. How did you assume the background air pollutants concentration level? In addition, how did you consider temporal and spatial distribution of air pollutants concentration level? In Beijing, can you ignore the traffic congestion?
5. I could not catch the energy sources of NEVs. In case of EVs, how did you consider the electricity generations and their impacts on health. In case of fuel-cell vehicles, how did you consider the hydrogen production and its impacts?
6. I could not catch the nember of NEVs sales if there are no subsidy. Even though, there are no subsidy, people may buy the NEVs.
Author Response
Dear Reviewer:
Thank you for reading our manuscript and reviewing it, which helped us improve it to a better scientific level. Those comments are all valuable and very helpful for revising and improving our paper, as well as the important guiding significance to our researches. We have carefully revised our manuscript and rechecked the entire text for errors in English spelling and grammar. The changes were marked up using the “Track Changes” function in the revised manuscript and highlighted in red in the response letter. We hope that the revision is acceptable and look forward to hearing from you soon.
Once again, thank you very much for your detailed comments and suggestions. Should there been any other corrections we could make, please feel free to contact us.
By the way, the COVID-19 pandemic is directly or indirectly affecting each one of us as members of the global community. We hope you and your loved ones continue to take the necessary precautions and stay safe and healthy.
Best Regards,
Yue Wang, Yang Wen, Yingying Xu, Lei Shi and Xuan Yang*
2022/10/17
Response to Reviewer 3 Comments
Point 1: Please provide current health costs (and/or DALY) by conventional vehicles and the share of mobile sources (compared with fixed sources).
Response 1: Thank you very much for your suggestion. Actually, fuel-cell vehicles had not been popularized in Beijing during the time scope of this study. In reality, Beijing's first hydrogen passenger car appeared on the market in 2022, and the vast majority of NEVs in Beijing are electric. Since an electric has no engine, they emit almost no pollutants. Therefore, the paper assumed that NEVs in Beijing do not discharge air pollutants. Pollutants emissions would be avoided if a consumer choose to buy a NEV instead of a conventional vehicle. In other words, the health costs due to conventional vehicle emission would be avoided by the NEVs consumption. In this paper, the avoided health costs by NEVs consumption are considered as equivalent to the health benefits of NEVs. That is the reason why we did not discuss the health costs by conventional vehicles specially. Thank you very much for the reasonable suggestion, we have realized that the importance of focusing on mobile sources should be introduced clearer in the manuscript and we added the description in line 32-36 of the manuscript, as follows.
In China, total motor vehicle emissions containing particulate matter, total hydrocarbon and other pollutants exceed approximately 44 million tons per year [2]. Local source emissions accounted for two-thirds of Beijing’s annual primary sources of PM2.5, while mobile source emissions contributed the most toward local source emissions in 2017 [3]. These emissions are the main causes of haze and photochemical smog pollution [4], which seriously threatens the atmospheric environment and human health [5][6].
References:
Zhang, L., Wang, L., Chai, J. Influence of new energy vehicle subsidy policy on emission reduction of atmospheric pollutants: A case study of Beijing, China. J. Clean. Prod. 2020, 275: 124069.
Lin, B., Tan, R. Estimation of the environmental values of electric vehicles in Chinese cities. Energy Pol. 2017, 104: 221-229.
Dong, K., Zeng, X. Public willingness to pay for urban smog mitigation and its determinants: a case study of Beijing, China. Atmos. Environ. 2018, 173: 355-363.
Point 2: Please add references related to cost and benefit analysis of transport negative externalities in the US and Europe.
Response 2: Thank you very much for your valuable comments. We approve that it is necessary to provide the relevant references which focused on the cost and benefit of transportation sector in the US and Europe. Your suggestions are very helpful for us to improve the reliability of this study. Therefore, we supplemented some references related to cost and benefit analysis of transportation sector’s negative externalities in line 92-108 and the relevant statements are as follows.
Moreover, adding the comparison of health effect attributes and policy costs seems to provide a wider context of the assessed policies [22-24]. Based on the case study of Thessaloniki, Sarigiannis et al. (2017) computed the potential co-benefits from reduced greenhouse gas emissions on public health by the year 2020 utilizing state-of-the-art concentration response functions for particulate matter, NO2 and C6H6. Results showed that the promotion of electric vehicles will provide monetary savings from the reduction in PM10, PM2.5, NO2 and C6H6 exposure up to 60.4, 49.1, 41.2 and 1.08 million Euros [25]. Raifman et al. (2021) analyzed all 378 counties in the Transportation Climate Initiative (TCI) region and found that even for the scenario with the smallest investment in active mobility, when it is fully implemented, TCI would result in hundreds of fewer deaths per year across the region, with monetized benefits in the billions of dollars annually [26]. Coomes et al. (2022) estimated benefits across a suite of child health outcomes in 42 New York City neighborhoods under the proposed regional Transportation and Climate Initiative. Results showed that a cap-and-invest strategy to reduce carbon emissions from the transportation sector could provide substantial health and monetized benefits to children in New York City through reductions in criteria pollutant concentrations, and with greater benefits among Black and Hispanic children [27].
References:
Rothengatter, W. Economic Valuation of Health Impacts in Cost-Benefit Analyses of Transport Infrastructure Projects in Europe. Encyclopedia of Environmental Health (Second Edition), 2019, 231-240.
Anenberg, S.C., Miller, J., Henze, D.K., Minjares, R., Achakulwisut, P. The global burden of transportation tail-pipe emissions on air pollution-related mortality in 2010 and 2015. Environ. Res. Lett. 2019, 14: 094012.
Arter, C., Buonocore, J., Chang, C., Arunachalam, S. Mortality-based damages per ton due to the on-road mobile sector in the Northeastern and Mid-Atlantic U.S. by region, vehicle class and precursor. Environ. Res. Lett. 2021, 16: 065008.
Sarigiannis, D.A., Kontoroupis, P., Nikolaki, S., Gotti, A., Chapizanis, D., Karakitsios, S. Benefits on public health from transport-related greenhouse gas mitigation policies in Southeastern European cities. Sci. Total Environ. 2017, 579: 1427-1438.
Raifman, M., Lambert, K.F., Levy, J.I., Kinney, P.L. Mortality Implications of Increased Active Mobility for a Proposed Regional Transportation Emission Cap-and-Invest Program. J. Urban Health 2021, 98: 315-327.
Coomes, K.E., Buonocore, J.J., Levy, J.I., Arter, C., Arunachalam, S., Buckley, L., Berberian, A., Gunasti, J., Perera, F. Assessment of the health benefits to children of a transportation climate policy in New York City. Environ. Res. 2022, 114165.
Point 3: Please provide the travel distance and emission factors by vehicle type. How did you consider trucks and lollys? In addition, these are affected by engine size, fuel type, speed and ages are quite important factors. How did you considered in the analysis?
Response 3: Thank you very much for the professional suggestion. Thanks to your first reminding about other types of vehicles, we realized that we did not define the range of our study. We only considered light-duty gasoline-fueled passenger vehicles whose engine displacement are greater than 1.0 Litre in Beijing, because this kind of vehicle is the most common in Beijing. As shown in 2020 Beijing Transport Development Annual Report, the ownership of light-duty gasoline-fueled passenger vehicles (LDGVs) account for 85.8% of the ownership of total motor vehicles. Actually, other types of motor vehicles are rarely seen in study region of this manuscript in 2019. According to “Notice on Traffic Management Measures to Reduce Pollutant Emissions for Some Trucks” released by Beijing Municipal Commission of Transport, Beijing Municipal Ecology and Environment Bureau and Beijing Traffic Management Bureau, from 06:00 to 23:00 each day, the roads within the Fifth Ring Road, which is the core area of Beijing and our study region, are prohibited for the passage of trucks, and the main roads of the Fifth Ring Road are prohibited for the ones with an approved load capacity of 8 tons (including) or more. Besides, the content of this regulation also includes that the roads within the Sixth Ring Road, which is the study region of this research and the main area of Beijing, are prohibited for the passage of all national â…¢ emission standard diesel trucks. Another reason we did not consider other types of vehicles was that travel distance and emission factors of trucks and lollys are not available in publicly data of Beijing. Thanks to your suggestion, we think it is necessary to consider as more types of vehicles as possible, and we hope that the data will be made public and our research can be improved in the future. We have added the explanation in line 127-132 of the manuscript and checked the whole manuscript in case any other definitions were missed. The detail of our supplement is as follows, thank you again for your reminding.
The light-duty gasoline-fueled passenger vehicles (LDGVs) is the most common type of vehicle in Beijing. According to Beijing Transport Development Annual Report in 2020, the ownership of LDGVs in 2019 account for 85.8% of the ownership of total motor vehicles. Besides, for the lack of travel distance and emission factors information of other types of vehicles in publicly data of Beijing, the LDGVs were chosen as the research object in this study.
For the second suggestion about the emission factors used in this manuscript, we think these factors are exactly very important. So, we rechecked the method we referred to and explained it clearer in line 144-149 in manuscript. The emissions factors were estimated by applying a new emission factor model for the Beijing vehicle fleet (EMBEV). The estimation applied a vehicle chassis dynamometer to measure emissions of gaseous pollutants over various test cycles. The test cycles included both the legislative driving cycle (i.e., New European Driving Cycle (NEDC) ) and alternative driving cycles with more real-world features (e.g., a typical transient driving cycle for LDGVs in Beijing) to justify the discrepancy between on-road emissions and type-approval test results. Emission measurement results of LDGVs complying with emission standards from Euro 1 to Euro 4 were collected for model development, including 1417 new LDGVs over NEDC, 217 in-use LDGVs over Beijing Driving Cycle (BJDC), 110 in-use LDGVs over both NEDC and BJDC, and 46 in-use LDGVs over the suite of speed-specific driving cycles. The calculation methodology for emission factors of LDGV technology groups is developed similarly to the MOBILE model (U.S. EPA, 2003[1]) and the COPERT model (Ntziachristos and Samaras, 2010[2]). The total emission factor for a defined LDGV group is the sum of emission factors in hot running, cold start and total hydrocarbons evaporation process. Emission factors in the hot running process further depend on a variety of parameters, including the vehicle technology group, accumulated vehicle kilometers traveled, average speed, fuel quality, and environmental conditions. The detailed introduction added in the manuscript in line 144-149 is as follows. Many thanks again for your professional suggestion.
To evaluate the differences between real emissions and the test results of approval, the estimation of emissions factors considered various test cycles including the legislative driving cycle and alternative driving cycles with more real-world features. The estimation also considered influence from the vehicle technology group, accumulated vehicle kilometers traveled, average speed, fuel quality, and environmental conditions.
Point 4: Not only mobile but also fixed emission sources affects the air pollution level. How did you assume the background air pollutants concentration level? In addition, how did you consider temporal and spatial distribution of air pollutants concentration level? In Beijing, can you ignore the traffic congestion?
Response 4: Thank you very much for the professional suggestion. For the first suggestion about background and fixed emission sources, we all think they are exactly important parameters. To avoid the influence of background environmental pollutants on the research results, we only considered the decrease of PM2.5 concentration driven by NEVs promotion, which was considered as equal to the emissions of the gasoline vehicles (GVs) they replace. Since the concentration of PM2.5 can be greatly affected by the large difference in climatic conditions between heating season and non-heating season in Beijing, the two periods were considered separately. According to existing results, the average height of the static weather boundary in heating season and non-heating season in Beijing is 420 m and 600 m, respectively. This paper focused on vehicle exhaust within the sixth ring road of Beijing, the area of this range is 2,267 km2. Hence, under the static stability weather, Beijing can be approximately regarded as a box with volume of 9.52 ×1011 m3 in heating season and 1.36 ×1012 m3 in non-heating season. As for the fixed emission sources, actually, the proportion of stationary sources in Beijing had been reduced to a very low level during our study period from 2016 to 2019, and existing stationary sources are generally clean and low-emission. According to a publicly released information by Beijing Ecological and Environmental Monitoring Center, motor vehicles are the largest source of pollution in Beijing from 2017 to 2020, which accounts for nearly a half of the total pollutant emission, while fixed emission sources is the least accounting for [10%, 12%]. Besides, Beijing has little local energy supply and relies mainly on energy from other provinces. As for the primary energy source, Beijing imported [97.99%, 98.95%] of electricity from other provinces according to the Energy Balance Table of Beijing from 2016 to 2019. In 2013, the government in Beijing had released a work plan for accelerating coal reduction and clean energy construction in Beijing from 2013 to 2017, which proposed a complete shutdown of Beijing's coal-fired power plants before 2017. In early 2017, the last large coal-fired power plant in Beijing was shut down, the goal of coal-free power plant has been realized in Beijing. We all agree that this suggestion is very important and briefly added the above introduction in line 201-207 the manuscript as follows.
According to a publicly released information by Atmospheric Department of Beijing Environmental Protection Bureau, motor vehicles are the largest source of pollution in Beijing. In early 2017, the last large coal-fired power plant in Beijing was shut down, the goal of coal-free power plant has been basically realized in Beijing during the study period from 2016 to 2019. Therefore, the proportion of stationary sources in Beijing had been reduced to a very low level and has less effect on the background concentration of PM2.5. For the geographical scope of this research is barely large enough to be significantly different in spatial parameters, this study assumed that the climatic and meteorological conditions and other spatial parameters in this area are the same. While the concentration of PM2.5 can be greatly affected by the large difference in climatic conditions between heating season and non-heating season in Beijing, the two periods were considered separately. According to existing results, the average height of the static weather boundary in heating season and non-heating season in Beijing is 420 m and 600 m, respectively. This paper focused on vehicle exhaust within the sixth ring road of Beijing, the area of this range is 2,267 km2. Hence, under the static stability weather, Beijing can be approximately regarded as a box with volume of 9.52 ×1011 m3 in heating season and 1.36 ×1012 m3 in non-heating season.
As for the spatial distribution of air pollutants concentration level, we all think it is an exactly important point to be considered. Actually, the geographical scope of our study is the area within the Sixth Ring Road of Beijing, which is barely large enough to be significantly different in spatial parameters. Therefore, we assumed that the climatic and meteorological conditions and other spatial parameters in this area are the same in our study. Thanks to your suggestion, we realized this assumption is very important for this manuscript but we did not introduce it before. So, we added the assumption in line 207-209 of the manuscript to improve the scientificity, the content is as follows.
For the geographical scope of this research is barely large enough to be significantly different in spatial parameters, this study assumed that the climatic and meteorological conditions and other spatial parameters in this area are the same. While the concentration of PM2.5 can be greatly affected by the large difference in climatic conditions between heating season and non-heating season in Beijing, the two periods were considered separately.
Regarding to the temporal distribution, we exactly think the concentration of PM2.5 can be greatly affected by the large difference in climatic conditions between heating season and non-heating season in Beijing. So, the two periods were considered separately when estimating the decrease of PM2.5 concentration driven by NEVs promotion and the economic effects of NEVs promotion. The concentration of PM2.5 in heating season is relatively high, so it was taken as the limes superior, and that in non-heating season was considered as the limes inferior, so as to get the estimation interval of the effects. The detailed content in line 239-241 is as follows. Thank you again for this suggestion.
The concentration of PM2.5 in heating season is relatively high, so it was taken as the limes superior, and that in non-heating season was considered as the limes inferior, so as to get the estimation interval of the effects.
Finally, the traffic congestion is exactly a very important problem in Beijing, thank you for this reasonable suggestion. Actually, the average speed of vehicles which is closely related to the traffic congestion was considered carefully when estimating the emission factors. Thanks to your suggestion, we have added the introduction of emission factors estimation in line 144-149 of the manuscript, which significantly improved the scientificity of the manuscript. The details are as follows. Many thanks again for these professional suggestions which improved this manuscript a lot.
To evaluate the differences between real emissions and the test results of approval, the estimation of emissions factors considered various test cycles including the legislative driving cycle and alternative driving cycles with more real-world features. The estimation also considered influence from the vehicle technology group, accumulated vehicle kilometers traveled, average speed, fuel quality, and environmental conditions.
Point 5: I could not catch the energy sources of NEVs. In case of EVs, how did you consider the electricity generations and their impacts on health. In case of fuel-cell vehicles, how did you consider the hydrogen production and its impacts?
Response 5: Thank you very much for the professional suggestion. Analyzing the whole life cycle of NEVs energy is indeed a very good point. We limited the scope of our study to Beijing while Beijing has little local energy supply and relies mainly on energy from other provinces. As for the primary energy source, Beijing imported [97.99%, 98.95%] of electricity from other provinces according to the Energy Balance Table of Beijing from 2016 to 2019. Therefore, most of Beijing's motor vehicle energy source emissions are in other provinces, which has very little impact on the health of local people in Beijing. That is the main reason why we did not take energy source emissions into account when measuring health benefits in Beijing. We added the reason in line 241-246 of the manuscript, the detailed content is as follows. As for the fuel-cell vehicles, actually, this types of NEVs had not been popularized in Beijing during the time scope of this study. In reality, Beijing's first hydrogen passenger car appeared on the market in 2022 and there were almost no hydrogen refueling stations in Beijing from 2016 to 2019. At the same time, the data of energy supply for NEVs is not publicly available and bias might be produced when separating the data related to NEVs from the macroscopic public energy data. We all hope the development of fuel-cell vehicles in Beijing can be faster, and we can get more detailed data of energy sources of NEVs in the future to improve our research. Many thanks again for your insightful suggestion.
This paper did not take energy source emissions into account when measuring health benefits in Beijing. According to the Energy Balance Table of Beijing from 2016 to 2019, almost [97.55%, 100%] of coal, 100% of natural gas, 100% of heat and [97.99%,98.95%] of electricity in Beijing were imported from other provinces. Therefore, most of Beijing's motor vehicle energy source emissions are in other provinces, which has very little impact on the health of people living in Beijing.
Point 6: I could not catch the nember of NEVs sales if there are no subsidy. Even though, there are no subsidy, people may buy the NEVs.
Response 6: Thank you very much for your professional suggestion. The purpose of this study is to quantify the economic benefits of NEVs promotion and compare them with the cost of promotion. It is difficult to quantify the cost of the government to promote the sales of NEVs, so this paper used subsidies to represent the cost. So, subsidies were not considered as a factor in this study but a measurement of cost. Obviously, subsidies cannot represent the full cost, and the cost were underestimated in this study. For this point is very reasonable, we added it in line 304-305 of the manuscript as follows, and hope to find a convictive way to measure the full cost in the future. At present, Beijing's subsidy policy is still in progress and may be cancelled in the future. Thanks to your valuable suggestion, we realized it is worthwhile to study the situation after abolishing the subsidy. Therefore, we will conduct comparative analysis after the subsidy policy is cancelled to expand our research in the future. Many thanks again for your insightful suggestions.
Since subsidies cannot represent the full cost of NEVs promotion by government, the cost is underestimated in this study.
Special thanks for these constructive comments!
[1] U.S. EPA, 2003. User’s guide to MOBILE6.1 and MOBILE 6.2, EPA420-R-03-110. http://www.epa.gov/otaq/models/mobile6/420r03010.pdf.
[2] Ntziachristos, L., Samaras, Z., 2010. EMEP/EEA emission inventory guidebook 2009. http://eea.europa.eu/emep-eea-guidebook (updated)
Round 2
Reviewer 1 Report
The article is now ready to be published.
Author Response
Thanks very much for your kind work and consideration on publication of our paper. We sincerely appreciate your insightful suggestions which will greatly help us revise and improve our study as well as provide valuable direction for future research.. On behalf of my co-authors, we would like to express our great appreciation to you.
By the way, the COVID-19 pandemic is directly or indirectly affecting each one of us as members of the global community. We hope you and your loved ones continue to take the necessary precautions and stay safe and healthy.
Thank you and best regards.
Yue Wang, Yang Wen, Yingying Xu, Lei Shi and Xuan Yang*
2022/10/20
Reviewer 3 Report
Please provide air pollutants share by vehicle type (in Beijing or other Chinese cities). We know that not gasoline-powered car but diesel-powered trucks have a high percentages.
https://www.statista.com/chart/16227/health-non-health-related-costs-of-air-pollution-from-transport-in-europe/
Please add the following papers.
https://www.pnas.org/doi/10.1073/pnas.1907956116
https://acp.copernicus.org/articles/19/6125/2019/
I believe negative impacts by electricity generation outside the Beijing should be also included for the analysis.
Author Response
Dear Reviewer:
Thank you for reading our manuscript and reviewing it, which helped us improve it to a better scientific level. Those comments are all valuable and very helpful for revising and improving our paper. We have carefully revised our manuscript and the changes were marked up using the “Track Changes” function in the revised manuscript and highlighted in red in the response letter. We hope that the revision is acceptable and look forward to hearing from you soon.
By the way, the COVID-19 pandemic is directly or indirectly affecting each one of us as members of the global community. We hope you and your loved ones continue to take the necessary precautions and stay safe and healthy.
Best Regards,
Yue Wang, Yang Wen, Yingying Xu, Lei Shi and Xuan Yang*
2022/10/20
Response to Reviewer 3 Comments
Point 1: Please provide air pollutants share by vehicle type (in Beijing or other Chinese cities). We know that not gasoline-powered car but diesel-powered trucks have a high percentages.
Response 1: Thanks for your valuable comments and suggestions. It is very useful for improving the scientificity of our research. We supplemented the air pollutants share by different types of vehicles and fuels in line 34-42 in accordance with your insightful recommendations, and the pertinent statements are as follows. Thanks again for your reasonable suggestions.
Local source emissions accounted for two-thirds of Beijing’s annual primary sources of PM2.5, while mobile source emissions contributed the most toward local source emissions in 2017 [3]. By types of vehicles, bus particulate matter emissions made up 9.9% of all vehicle emissions in 2019, while truck particulate matter emissions totaled 62,000 tons, or 90.1% of all vehicle emissions. For different types of fuels, gasoline cars produced 5.576 million, 1.328 million and 0.303 million tons of CO, HC and NOx emissions in 2019, respectively, accounting for 80.3%, 77.5% and 4.9% of all vehicle emissions nationally. Diesel vehicle emissions of the aforementioned pollutants were 1.286 million, 0.213 million, 5.532 million and 0.069 million tons, or 18.5%, 12.5%, 88.9% and 99%, respectively. Gas vehicle emissions of CO, HC and NOx were 81,000, 171,000, and 387,000 tons, or 1.2%, 10.0%, and 6.2%, respectively [4]. These emissions are the main causes of haze and photochemical smog pollution [5], which seriously threaten the atmospheric environment and human health [6][7].
References:
Ministry of Ecology and Environment of the People’s Republic of China. China Mobile Source Environmental Management Annual Report (2020). Available online: https://www.mee.gov.cn/hjzl/sthjzk/ydyhjgl/202008/P020200811521365906550.pdf (accessed on 19 October 2022)
Point 2: Please add the following papers.
Response 2: We sincerely appreciate your insightful suggestions of adding some authoritative literature to increase the readability of this article. We supplemented the related references and the details are as follows. Thanks again for your invaluable suggestion to make our description more accurate and scientific.
The light-duty gasoline-fueled passenger vehicles (LDGVs) are the most common type of vehicle in Beijing. All out-of-city diesel vehicles with emissions standards lower than China III were prohibited from travelling within the sixth ring road as of September 2017 [31].
The exposure–response model of PM2.5 for each health endpoint, which were derived from available epidemiologic studies and based on a relative risk model in the form of Poisson regression, has become a mature theory for measuring health effects of pollution [38] and were used to quantify the health effects of particulate air pollution [39-42].
References:
Cheng, J., Su, J., Cui, T., Li, X., Dong, X., Sun, F., Yang, Y., Tong, D., Zheng, Y., Li, Y., Li, J., Zhang, Q., He, K. Dominant role of emission reduction in PM2.5 air quality improvement in Beijing during 2013-2017: a model-based decomposition analysis. Atmos. Chem. Phys. 2019, 19: 6125-6146.
Zhang, Q., Zheng, Y., Tong, D., Shao, M., Wang, S., Zhang, Y., Xu, X., Wang, J., He, H., Liu, W., Ding, Y., Lei, Y., Li, J., Wang, Z., Zhang, X., Wang, Y., Cheng, J., Liu, Y., Shi, Q., Yan, L., Geng, G., Hong, C., Li, M., Liu, F., Zheng, B., Cao, J., Ding, A., Gao, J., Fu, Q., Huo, J., Liu, B., Liu, Z., Yang, F., He, K., Hao, J. Drivers of improved PM2.5 air quality in China from 2013 to 2017. Proc. Natl. Acad. Sci. 2019, 116: 24463-24469.
Point 3: I believe negative impacts by electricity generation outside the Beijing should be also included for the analysis.
Response 3: Many thanks for your professional suggestion. We all agree that this is a very valuable opinion which can contribute to a more comprehensive assessment of the effectiveness of the policy. However, electricity in Beijing is mainly imported from other regions through the nationwide power grid. As for the primary energy source, Beijing imported [97.99%, 98.95%] of electricity from other provinces according to the Energy Balance Table of Beijing from 2016 to 2019. To our deep regret, there are almost no publicly available data on which provinces exactly the electricity is coming from, and almost no data on what type of electricity is being delivered to Beijing or the proportions of different types of electricity (coal, hydro, solar, wind, etc.), so it is difficult to accurately quantify the negative impacts by electricity generation outside the Beijing. However, the question you have raised indicates a significant direction for further expansion of this study in the future. We all agree that this direction is important and need to be told to the readers. So, we specially added a supplement part at line 331-343 of the manuscript to explain this point as you suggested, we hope the detailed data can be available and this research can be more sophisticated in the future. The added part of this manuscript is as follows. Thank you very much again for this insightful suggestion.
3.4. Limitations
This study limited the scope to Beijing who has little local energy supply and relies mainly on energy from other provinces. As for the primary energy source, Beijing imported [97.99%, 98.95%] of electricity from other provinces according to the Energy Balance Table of Beijing from 2016 to 2019. Therefore, most of Beijing's motor vehicle energy source emissions are in other provinces. Although these emissions have very little impact on the health of local people in Beijing, provinces that supply electricity to Beijing may face local air pollution problems. However, due to the lack of public data, it is difficult to identify the provinces that supply electricity to Beijing. The types of electricity being delivered to Beijing and the proportions of different types of electricity (coal, hydro, solar, wind, etc.) are hard to be identified as well, which limits further refinement of the study. With the continuous refinement of public data, the negative impacts by electricity generation outside the Beijing can be considered in the future.
Special thanks again for your patient review and constructive suggestions!!